# Wavelet Coherency Structure in Open Channel Flow

**Kebing Chen [1], Yifan Zhang [1] and Qiang Zhong [1,2,\*]**

[1]  College of Water Resources and Civil Engineering, China Agricultural University, Beijing 100083, China
[2]  Beijing Engineering Research Center of Safety and Energy Saving Technology for Water Supply Network System, China Agricultural University, Beijing 100083, China
\*  Correspondence: qzhong@cau.edu.cn

**Abstract:** Many studies based on single-point measurement data have demonstrated the impressive ability of wavelet coherency analysis to catch the coherent structures in the wall-bounded flows; however, the question of how the events found by the wavelet coherency analysis relate to the features of the coherent structures remains open. Time series of velocity fields in $x$–$y$ plane of the steady open channel flow was obtained from a time-resolved particle imaging velocimetry system. The local wavelet spectrum found shows that one of the main energetic scales in open channel flows is $3h$ motions. The wavelet coherent coefficients of $u$ and $v$ series from the same measurement points successfully detected the occurrence and the scale of these $3h$ motions, and the phase angle indicates their inside velocity structure is organized by the Q2 and Q4 events. The wavelet coherency analysis between different measurement points further reveals the incline feature of the $3h$ scale motions. All the features of this $3h$ motion found by the wavelet coherency analysis coincide well with the flow field that is created by the passing of hairpin packets.

**Keywords:** wavelet coherency; Taylor's frozen turbulence hypothesis; scale; hairpin vortex packet; open channel flow

## 1. Introduction

The essential difficulties of turbulence are the numerous scale motions and the complex interactions between them. Therefore, the two major tasks of turbulent data analysis can be summarized: The first one is scale decomposition, which decomposes the different scale components from the original turbulent signal; the other one is the correlation of scales, which analyzes the interactions between different scales.

Traditional turbulent data analysis is based on the Fourier transform. Fourier analysis decomposes the turbulent signal into the sum of infinite sine and cosine functions with different frequencies. It can reveal the frequency characteristic of the turbulent signal, but it cannot determine when those frequency components occurred. This is due to the fundamental difference between the presuppositions of Fourier analysis and intermittency of the turbulent signal. Specifically, each frequency in Fourier analysis stands for one scale vortices, and the passing of one vortex will cause one period sine wave at the measurement point. The continuous sine waves from positive to negative infinity on the time axis make Fourier analysis implicitly presuppose that each scale vortices occupy the entire timeline. However, the vortices in turbulence occur intermittently, and they present in the portion of both time and space axis and only effect the local flow field.

Wavelet analysis overcomes this defect by using the waves fluctuating in the limited time as the basis functions, and the behavior of the signal at infinity does not play any role. Thus, wavelet analysis is performed in both time and frequency domain, allowing the definition of local spectral properties and the ability to zoom in on local features of the signal. Since Farge [1] introduced the wavelet analysis

into turbulence, wavelets have become pervasive in turbulent signal analysis. Most of the applications use the wavelet analysis only as a scale decomposition tool, and the research on the correlation of scales are still based on the traditional correlation coefficient or the Fourier coherency analysis.

Following the definition of coherency in Fourier analysis, Liu [2] firstly defined the wavelet coherency and applied the signal from ocean wind waves. The wavelet coherency analysis can reveal the similarity of two signals at different scales and instants, which provides a powerful tool to reveal the correlation between two signals of different types or from different locations which is cased by the local coherent structures. The fundamentals of the cross-wavelet and wavelet coherency analysis were systematically introduced by Torrence and Compo [3], and Torrence and Webster [4] proposed a broadly applicable smoothing function for the wavelet coherency, which made wavelet coherency eventually become a universal data analysis method. Grinsted et al. [5] developed a software package that allows users to perform the cross wavelet transform and wavelet coherency analysis, and they applied the methods to the Arctic Oscillation index and the Baltic maximum sea ice extent record. Camussi et al. [6] applied the cross wavelet transform and wavelet coherency analysis to wall pressure fluctuation signals from a microphone pair in the incompressible turbulent boundary layers. The time instants corresponding to a local wavelet coherency overcoming the trigger level were selected as the coherent events, and conditional statistical analysis was performed on these selected events to show the physical features of these highly coherent events. Based on the conditional statistics, the highly coherent event was speculated as the footprint of the near-wall-sweep-type motion which are known to be closely associated with the presence of streamwise vortices embedded within the turbulent flow and located in the near-wall region.

From the previous literature review, it can be seen that most of the applications of the wavelet coherency was done on single point measurement signals. These single point signals contain the information of the events that happen in turbulence, but they cannot reflect the whole picture of the events, and the question of how these highly coherent events revealed by the wavelet coherent analysis from single point measurement data related to the two or three dimensional coherent structures in the flow field remains open. Although we can easily get two- or even three-dimensional flow fields in the laboratory today, a large number of tasks in the actual application scenario are still completed by single-point measurement. Therefore, it is instructive for the single point data analysis to explore the three-dimensional coherent structures corresponding to events in them. Present study employs the time-resolved particle imaging velocimetry system (TR-PIV) to get the two dimensional flow field time series, which makes it possible to find out the flow structures from the 2D flow fields at the corresponding wavelet coherent events from the single point time series. The following part of this paper is organized as follows. Section 1 describes the principle of the wavelet coherency analysis briefly. Section 2 introduces the experiment and data. Section 3 presents the results of wavelet spectral and coherency analysis. Section 4 summarizes major findings.

## 2. Wavelet Coherency

We limit ourselves to a brief introduction of the wavelet coherency. The detailed procedures can be found in [3–5]. The continuous wavelet transform of a velocity signal $u(t)$ is defined as the convolution of $u(t)$ with a scaled and normalized wavelet:

$$W_u(a, t_0) = \int_{-\infty}^{+\infty} \left[ u(t') \frac{1}{\sqrt{a}} \psi^* \left( \frac{t_0 - t'}{a} \right) \right] dt' \tag{1}$$

where $\psi$ is the wavelet function; superscript * denotes the complex conjugate; $a$ and $t_0$ are the scale and position parameter, respectively; $W_u(a, t_0)$ is the wavelet transform coefficient of $u$ at scale $a$ and instant $t_0$. The wavelet function $\psi$ in the wavelet transform must have zero mean and be localized in both time and frequency domain [1]. Following Liu [2], Torrence and Compo [3] and Chui [7],

the complex Morlet wavelet is used in this study since it provides a good balance between time and frequency localization:

$$\psi(t) = \pi^{-1/4} e^{i\omega_0 t} e^{-\frac{t^2}{2}} \tag{2}$$

where $\omega_0$ is a wavelet center frequency, here taken to be 6 to balance between time and frequency localization. From Equation (1), it can be seen that the wavelet function is stretched and shifted in time by varying the scale and position parameter. Thus, different scale motions at every instant can be separate out from the original signal by the wavelet transform. In order to facilitate comparison, the pseudo Fourier frequency corresponding to the scale $a$ is usually determined by picking up the energy peak in the Fourier spectrum of the wavelet function. For the Morlet wavelet with $\omega_0 = 6$, the Fourier frequency $f$ is almost equal to the scale, e.g., Fourier frequency $f_0$ is 0.971Hz when $a = 1$, and the corresponding frequency $f$ at scale $a$ is determined by:

$$f = \frac{f_0}{a} \tag{3}$$

When the pseudo Fourier frequency $f$ is determined, the wave number $k$ and wavelength $\lambda$ can be obtained based on Taylor's hypothesis:

$$\begin{cases} k = \frac{2\pi f}{U} \\ \lambda = \frac{2\pi}{k} \end{cases} \tag{4}$$

where $U$ is the mean streamwise velocity at that point.

Similar to Fourier analysis, the cross wavelet coefficient can be obtained:

$$W_{uv}(a,t) = W_u(a,t) W_v^*(a,t) \tag{5}$$

There is a Parseval relation [8]:

$$\int_{-\infty}^{+\infty} [u(t)v^*(t)] dt = \frac{1}{c_\psi} \int_{-\infty}^{+\infty} \int_0^{+\infty} W_u(a,t) W_v^*(a,t) \frac{da}{a^2} dt = \frac{1}{c_\psi} \int_{-\infty}^{+\infty} \int_0^{+\infty} W_{uv}(a,t) \frac{da}{a^2} dt \tag{6}$$

where $c_\psi$ is a factor depended by the wavelet function:

$$\begin{cases} c_\psi = \int_0^\infty |\hat{\psi}(\omega)|^2 \frac{d\omega}{|\omega|} \\ \hat{\psi}(\omega) = \int_{-\infty}^{+\infty} \psi(t) e^{-2i\pi\omega t} dt \end{cases} \tag{7}$$

In the turbulence situation, the velocity $u(t)$ and $v(t)$ are both real signal. Thus, we have:

$$\begin{aligned} E_{total} = \int_{-\infty}^{+\infty} u(t)^2 dt &= \frac{1}{c_\psi} \int_{-\infty}^{+\infty} \int_0^{+\infty} |W_{uu}(a,t)|^2 \frac{da}{a^2} dt \\ &= \frac{1}{c_\psi k_0} \int_{-\infty}^{+\infty} \int_0^{+\infty} \left|W_{uu}\left(\frac{f_0}{f},t\right)\right|^2 df dt \end{aligned} \tag{8}$$

where $E_{total}$ can be considered as the turbulent kinetic energy. Equation (8) shows that the wavelet transform coefficients actually represent the energy carried by the corresponding scale motions at the given instant. Thus $|W_{uu}(f,t)|^2$ is called local wavelet spectrum. The frequency spectrum can be defined as:

$$E_{uu}(f) = \frac{1}{C} \int_{-\infty}^{+\infty} |W_{uu}(f,t)|^2 dt \tag{9}$$

where $C$ is the normalization constant which makes

$$\int_0^{+\infty} E_{uu}(f)df = u'^2 \tag{10}$$

in which $u'$ is the standard deviation of time series $u(t)$.

For the correlation coefficient, we have the definition:

$$R_{uv} = \frac{\int_{-\infty}^{+\infty}[u(t)v(t)]dt}{\sqrt{\int_{-\infty}^{+\infty}u(t)^2dt} \cdot \sqrt{\int_{-\infty}^{+\infty}v(t)^2dt}} \tag{11}$$

Based on Equation (6), we can get:

$$R_{uv}^2 = \frac{\Re\left[\int_{-\infty}^{+\infty}\int_0^{+\infty}W_{uv}(a,t)\frac{da}{a^2}dt\right]^2}{\left[\int_{-\infty}^{+\infty}\int_0^{+\infty}\left|W_u(a,t)\right|^2\frac{da}{a^2}dt\right] \cdot \left[\int_{-\infty}^{+\infty}\int_0^{+\infty}\left|W_v(a,t)\right|^2\frac{da}{a^2}dt\right]} \tag{12}$$

in which $\Re[\cdot]$ is the real part of complex variables. Similar to the Equation (12), we can define the wavelet coherency as

$$R_{uv}^2(a,t) = \frac{\Re[W_{uv}(a,t)]^2}{\left|W_u(a,t)\right|^2 \cdot \left|W_v(a,t)\right|^2} \tag{13}$$

However, the correlation between two signals at one instant obviously makes no sense. As noted by Liu [2], this coherency is identically one at all times and scales. This problem is circumvented by smoothing the wavelet coefficient along time or both time and scale axis before normalizing. The time and scale smoothing operator given by Torrence and Webster [4] is used in this study:

$$\begin{cases} R_{uv}^2(a,t) = \frac{\left|S\left[a^{-1}W_{uv}(a,t)\right]\right|^2}{S\left[a^{-1}\left|W_u(a,t)\right|^2\right] \cdot S\left[a^{-1}\left|W_v(a,t)\right|^2\right]} \\ S[W] = S_{scale}(S_{time}(W)) \\ S_{scale}(W(a,t)) = W(a,t) * c_1\Pi(0.6a) \\ S_{time}(W(a,t)) = W(a,t) * c_2 e^{\frac{-t^2}{2a^2}} \end{cases} \tag{14}$$

in which the symbol * denotes the convolution product; $\Pi(0.6a)$ is a boxcar filter of width 0.6; $e^{\frac{-t^2}{2a^2}}$ is the absolute value of the Morlet wavelet; $c_1$ and $c_2$ are normalization coefficients to have a total weight of unity. The factor of 0.6 is the empirically determined scale decorrelation length for the Morlet wavelet. The wavelet-coherence phase difference is given by:

$$\phi(a,t) = \arctan\left\{\frac{\Im\left[S\left[a^{-1}W_{uv}(a,t)\right]\right]}{\Re\left[S[a^{-1}W_{uv}(a,t)]\right]}\right\} \tag{15}$$

in which $\Im[\cdot]$ is the imaginary part of the complex variables.

The MATLAB function "wcoherence" is used to compute the wavelet coherency between two signals in this study. One can find the wavelet coherency of analytical signals in the documentation of the MATLAB function, which reveals the ability of wavelet coherency to analyze the scale and phase relations between two signals.

## 3. Experiment

Experiments were conducted in the Tsinghua tilting hydraulic flume, which is a closed-circuit open channel 20 m long and 0.3 m wide. The measurement section was set up 12 m downstream of the flume entrance. Eight ultrasonic water level sensors were set on the flume to monitor the water depth across the entire flume. The streamwise and wall-normal directions are denoted by $x$ and $y$, respectively, and the corresponding components of fluctuating velocities are $u$ and $v$. Velocity field measurements of the $x$–$y$ plane in the middle of channel were made at a uniform flow condition as listed in Table 1. The water depth $h$ is 2.9 cm. Thus, the flow in the central region can be considered as statistically two-dimensional [9,10]. The Reynolds number based on the section average velocity $U_m$ and water depth was 15,895 ($U_m$ was based on the discharge from the electromagnetic flowmeter), and the friction Reynolds number was 880 ($u_* = \sqrt{ghJ}$).

**Table 1.** Flow condition and particle imaging velocimetry (PIV) parameters.

| $J$ | $h$ | $v$ | $B/h$ | $U_m$ | $u^*$ | $Fr$ | $Re$ | $Re_\tau$ |
|---|---|---|---|---|---|---|---|---|
| - | (cm) | ($10^{-2}$ cm$^2$/s) | - | | (cm/s) | - | - | - |
| 0.0036 | 2.90 | 1.06 | 10.3 | 58.1 | 3.29 | 1.09 | 15,895 | 880 |

| Image size pixels | Exposure time μs | Frequency Hz | Resolution pixels/mm | Number of images |
|---|---|---|---|---|
| 1280 × 896 | 150 | 2500 | 32 | 5596 |

$J$ = bed slope, $h$ = water depth, $v$ = kinematic viscosity, $B$ = channel width, $B/h$ = aspect ratio, $U_m$ = the depth-averaged velocity, $u^*$ = friction velocity, $Fr$ = Froude number, $Re$ = Reynolds number, $Re_\tau$ = friction Reynolds number.

Instantaneous, two-dimensional velocity fields were measured in the streamwise-wall-normal plane ($x$–$y$ plane) with a TR-PIV. The laser sheet was projected from the channel bed and it was located at the central line of the channel. The camera was set at the side of the channel and the laser sheet plane and the CMOS plane of the camera kept parallel to the mid-vertical plane of the channel. The PIV parameters are also listed in the Table 1. The exposure time was fixed at 150 μs as a compromise between minimizing image streaking and maximizing image lightness. The sampling frequency was 2500 Hz to obtain time-resolved series of 2D flow fields. A total number of 5596 images were obtained thus the time series contains 5595 continuous velocity fields. A greater sample number will be better but the capacity of the high-speed memory in the high-speed camera limits the number of images. Particle images were analyzed by using the iterative multi-grid image deformation method. Various test results of the PIV algorithm used in this paper can be found in the report of the 4th International PIV Challenge (the symbol of the PIV algorithm is TsU) [11]. The window size in the final iterative step is 16 × 16 pixels with a 50% overlap. A detailed description of the experiment system can be found in [12,13].

## 4. Results

### 4.1. Wavelet Spectrum

Figure 1a presents the wavenumber spectrum $E_{uu}$ for streamwise velocity fluctuation at point ($x/h = 0$, $y/h = 0.5$) from both Fourier and wavelet analysis. The biggest wavelength was chosen as 10$h$ because the length of the time series is $tU/h = 45.5$, and this length is not enough for obtaining a creditable result for the larger scale motions. It can be seen from Figure 1a that the wavelet spectrum is much smoother and follow the Fourier spectrum well. This can be attributed to the fact that the wavelet spectrum presents an average of the Fourier spectrum weighted by the square of the Fourier transform of the analyzing wavelet shifted at wave number $k$, and it keeps the same power-law as in the Fourier spectrum [14]. Owing to the relatively small sample size, even the wavelet spectrum is still spiky. The TR-PIV system can capture the 2D flow field time series, but it is very hard to get big

sample sizes because of the storage and transport difficulties of the huge amount of data. However, the main goal of this study is to show the physical meaning revealed by the wavelet analysis on the turbulent signal from open channel flows instead of presenting accurate wavenumber spectra.

Figure 1b shows the pre-multiplied power spectrum, $kE_{uu}(k)$, at point ($x/h = 0$, $y/h = 0.5$) from the wavelet spectrum. Based on Equation (10), when the horizontal axis is set to logarithmic coordinates, we have

$$u'^2 = \int\limits_0^{+\infty} E_{uu}(k)dk = \int\limits_0^{+\infty} kE_{uu}(k)d(\ln k) \tag{16}$$

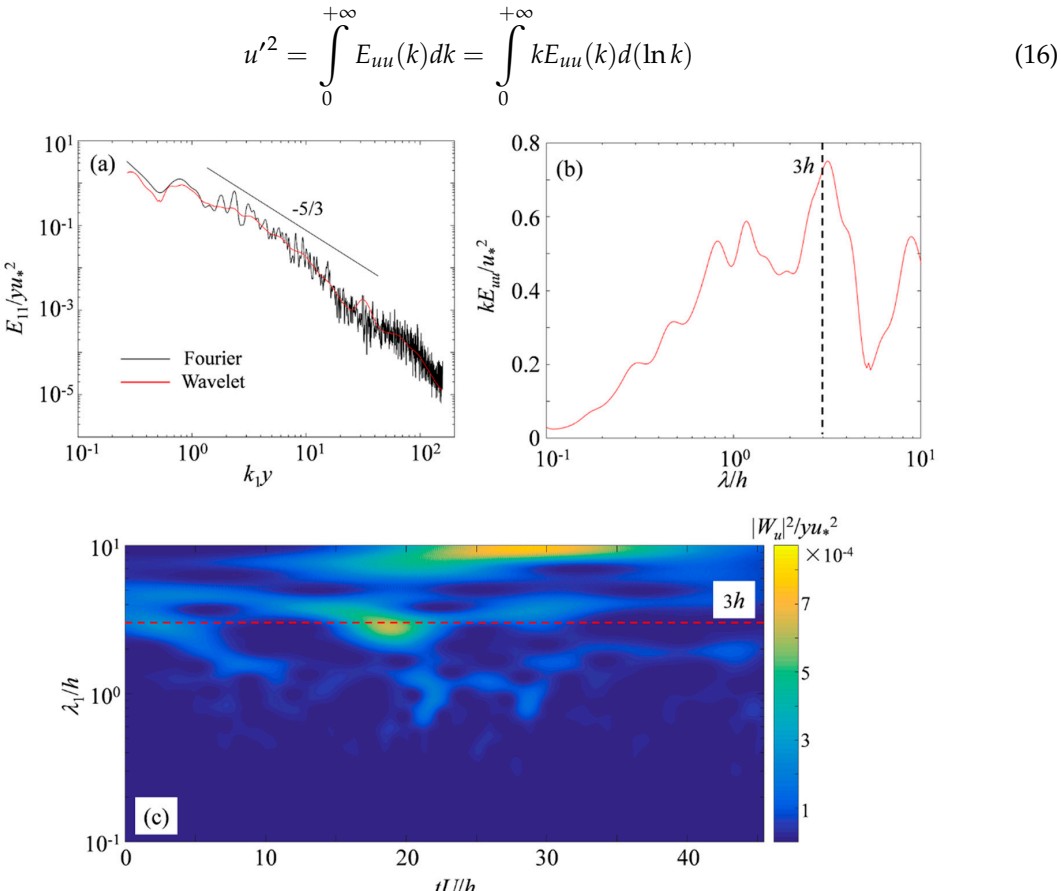

**Figure 1.** The wavenumber spectrum (**a**), pre-multiplied spectrum (**b**) and local wavelet spectrum (**c**) of the streamwise velocity series at the measurement point ($x/h = 0$, $y/h = 0.5$). The black solid straight line in (**a**) shows the classical $-5/3$ law in the turbulence. The black dash line in (**b**) and red dash line in (**c**) is marked the strongest scale in the spectrum.

Therefore, the whole area under the pre-multiplied spectrum curve in the semi-log plot is directly related to the value of turbulence intensity, and the area under a small section of the spectrum curve can be considered as the strength of the corresponding scale motions. The strongest scale in Figure 1b is approximately $3h$ as marked by the black dash line, which means the $3h$ scale motions are one of the main energetic structures in the outer layer. In the scales greater than $3h$, there may exist another energetic mode around $10h$. This two-energetic-mode feature of the pre-multiplied spectrum is similar to the results from other wall-bounded flows [15–17]. By the coherent structure classification in open channel flow [12,13], the $h$ and $10h$ order motions can be classified as large- and very-large-scale structures.

The traditional spectrum gives the general information about the energetic scales but cannot show the time instants when the local event happens. The contour map of the local wavelet spectrum, $|W_u|^2$ is shown in Figure 1c. There are two high energy regions at scale approximately $3h$ and $10h$, respectively, which coincides with the result in pre-multiplied spectrum in Figure 1b. From $|W_u|^2$, we can see that the strongest $3h$ and $10h$ scale motions pass the measurement point during $17 < tU/h < 21$ and $25 < tU/h < 35$, respectively.

In order to show the entire flow structures when the local wavelet spectrum shows high energy, fluctuating velocity field series from $tU/h = 11$ to 35 are pieced together based on Taylor's frozen turbulence hypothesis in Figure 2, following Zhong et al. [12].

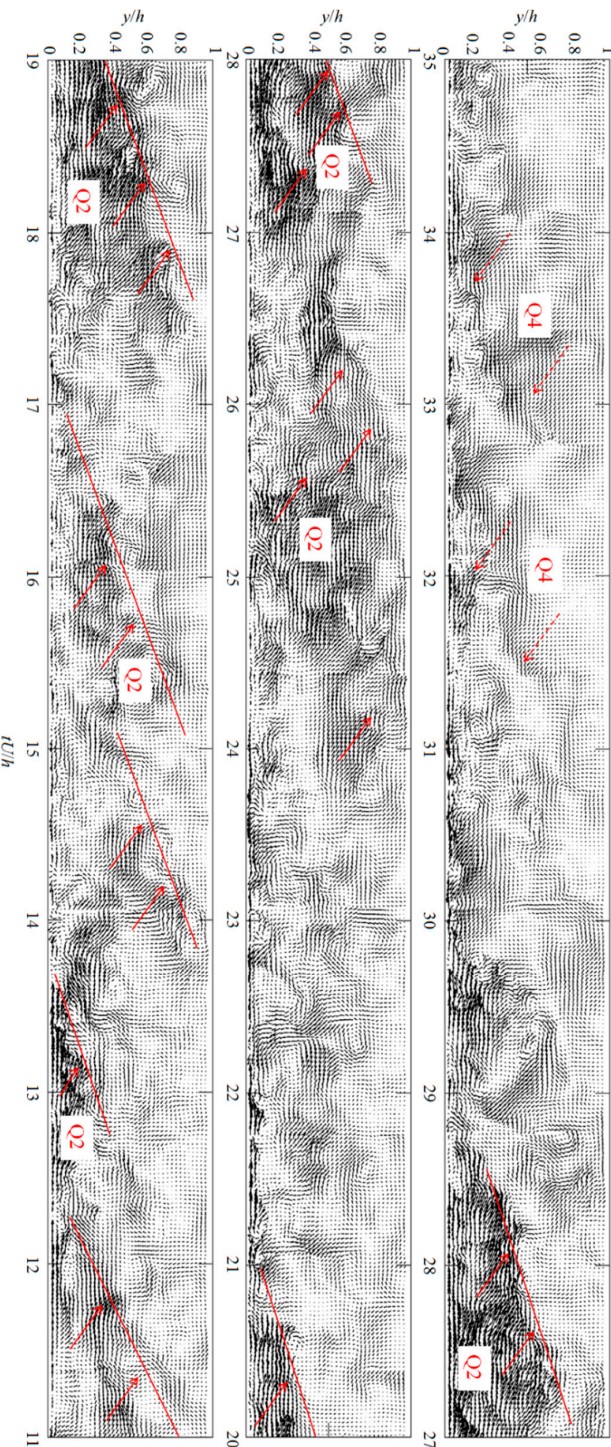

**Figure 2.** Fluctuating velocity fields pieced together based on Taylor's frozen turbulence hypothesis. The red arrows indicate the strong Q2 and Q4 events in the flow field, and the red solid lines indicate the inclined shear layers.

The convection velocity in Taylor's frozen turbulence hypothesis is the mean velocity of the whole field. The major characteristic of flow field during $17 < tU/h < 21$ is an inclined shear layer

(marked by the red solid lines in Figure 2) with the strong ejections (marked by the red solid arrows in Figure 2). The ejections are also called the Q2 event, where the fluctuating streamwise velocity $u < 0$ and vertical velocity $v > 0$. When $u > 0$ and vertical velocity $v < 0$, there is a sweep motion, or Q4 event. The streamwise scale of this inclined shear layer structure is approximately 2.5$h$. The features of this inclined shear layer structure are similar to the flow field in the reference [12,18–20], and it is usually considered as the typical sign of the passing of a hairpin packet [19,21]. The transport effect of the hairpin vortices in the packet induce the strong Q2 events, and the shear line is inclined because the heads of hairpin vortices in the packet describe an envelope inclined at 15–20° with respect to the wall. Considering the information from flow field and the results in previous literature, the highly energetic region at 3$h$ scale in the wavelet spectrum reveals the passing of the hairpin packet. In Figure 2 it can be seen that from $tU/h$ = 11 to 30 there are at least six typical inclined shear layers located almost end to end, and the wavelet spectrum indeed shows high energy in the corresponding duration and scale.

For the 10$h$ scale highly energetic region, the flow field during $25 < tU/h < 35$ shows hairpin packets firstly and then strong Q4 events (marked by the red dash arrows in Figure 2). It indicates that the 10$h$ order motions consist of two different types of smaller structures, hairpin packets, and Q4 events. This agrees with the phenomena reported by Zhong et al. [12], Adrian and Marusic [22], Zhong et al. [12] and Zhong et al. [13] suggested a super-streamwise vortex model for the 10$h$ order motions in open channel flows. The super-streamwise vortices rotate around the $x$ axis. The strong Q2 events from hairpin packets constitute the upward movement and the Q4 events are the downward movement of the super-streamwise vortices.

From the above discussion, the energetic scales can be presented by the traditional spectrum analysis, and the wavelet spectrum can show not only the energetic scales but also the moment these energetic structures occur. However, wavelet spectrum contains no information about the organization of these structures. The following analysis will show that the wavelet coherency can reveal more details about the energetic structures.

*4.2. Wavelet Coherency*

The wavelet coherency between time series $u$ and $v$ at the same measurement point ($x/h$ = 0, $y/h$ = 0.5) is shown in Figure 3. The advection velocity $U$ is the mean velocity of the whole field. The white dash line marks the cone of influence of the boundaries. The biggest wavelength was chosen as 10$h$ as in Figure 1. It can be seen from Figure 3 that the most remarkably coherence appears in $13 < tU/h < 32$ (marked by the white rectangle) near the scale 3$h$ (marked by the red dash line) and in $tU/h > 35$ at the scale 3$h$ to 10$h$. Most of the coherence area of $tU/h > 35$ is beyond the white dash line, which is the cone of influence for the wavelet, and the wavelet coherent value is untrusted because of too close to the beginning or ending instantaneous. Thus, we only focus on the $13 < tU/h < 32$ region. The phase angle in this area is about $\pi$. This highly coherent area indicates that there are 3$h$ scale structures continuously passing the measurement point during $13 < tU/h < 32$. From Figure 2, one can see that there are several hairpin packets in the flow field during $13 < tU/h < 32$. Thus, the highly coherent area near the scale $\lambda/h$ = 3 in Figure 3 reveals the passing of several hairpin packets. In addition, the $\pi$ phase angle further reveals the velocity structure inside the hairpin packets. As shown in Figure 4, when the hairpin packet passes the measurement point, the Q2 events lead to that the streamwise fluctuation $u$ appearing as positive firstly, and the vertical fluctuation $v$ appearing as negative. Both $u$ and $v$ cross 0 when the shear layer passes the measurement point, and then $u$ and $v$ turn into negative and positive, respectively. Therefore, the phase difference between time series $u$ and $v$ has to be approximately 180° when hairpin packets are passing. The $\pi$ phase angle reveals the feature of the hairpin packets that the Q2 and Q4 events dominate their inside velocity structures.

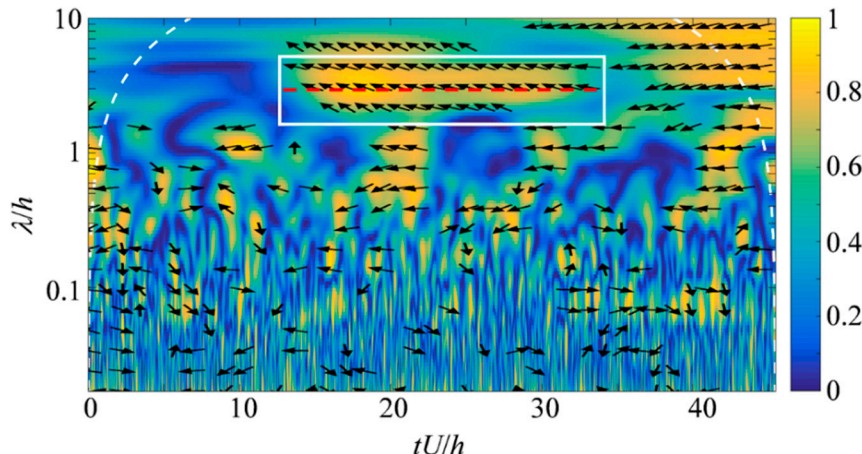

**Figure 3.** Wavelet coherent coefficients between the streamwise and vertical velocity series at the measurement point ($x/h = 0$, $y/h = 0.5$). The white box marks the time duration as the same as that in Figure 2, and the red dash line indicates the $3h$ scale. The high coherent area appears in the white box and the phase difference is approximately $\pi$.

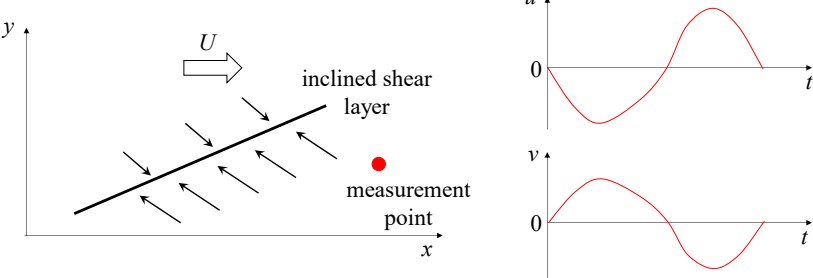

**Figure 4.** The explanation of the phase angle in the wavelet coherency between the streamwise and vertical velocity series at the same measurement point when the incline structures pass. When the hairpin packet passes the measurement point, the Q2 events lead to that the streamwise fluctuation $u$ appears in positive firstly, and the vertical fluctuation $v$ appears in negative. After the inclined shear layer passes the measurement point, $u$ and $v$ turn into negative and positive, respectively.

The $u$ series at ($x/h = 0$, $y/h = 0.5$) was chosen as the fixed point to do the wavelet coherency with other three measurement points. Figure 5 shows the wavelet coherency of $u$ series at ($x/h = −0.35$, $y/h = 0.15$), ($x/h = 0$, $y/h = 0.15$) and ($x/h = 0.35$, $y/h = 0.15$), respectively. Points ($x/h = −0.35$, $y/h = 0.15$) and ($x/h = 0.35$, $y/h = 0.15$) are located at the upstream and downstream of point ($x/h = 0$, $y/h = 0.5$), respectively, and ($x/h = 0$, $y/h = 0.15$) is directly below the fixed point. From Figure 5a, there are two highly coherent areas during $11 < tU/h = 35$ near the scale $3h$. The first one is roughly from $tU/h = 11$ to 23 (marked by the white box). The second one is approximately from $tU/h = 27$ to 30 and much weaker than the correlation during the same duration in Figure 3, which can be attributed to the fact that the correlation reduces when the distance between two measurement points increases. Referring to Figure 2, these two highly coherent areas are related to the typical hairpin packets during $11 < tU/h < 21$ and $27 < tU/h < 29$, respectively.

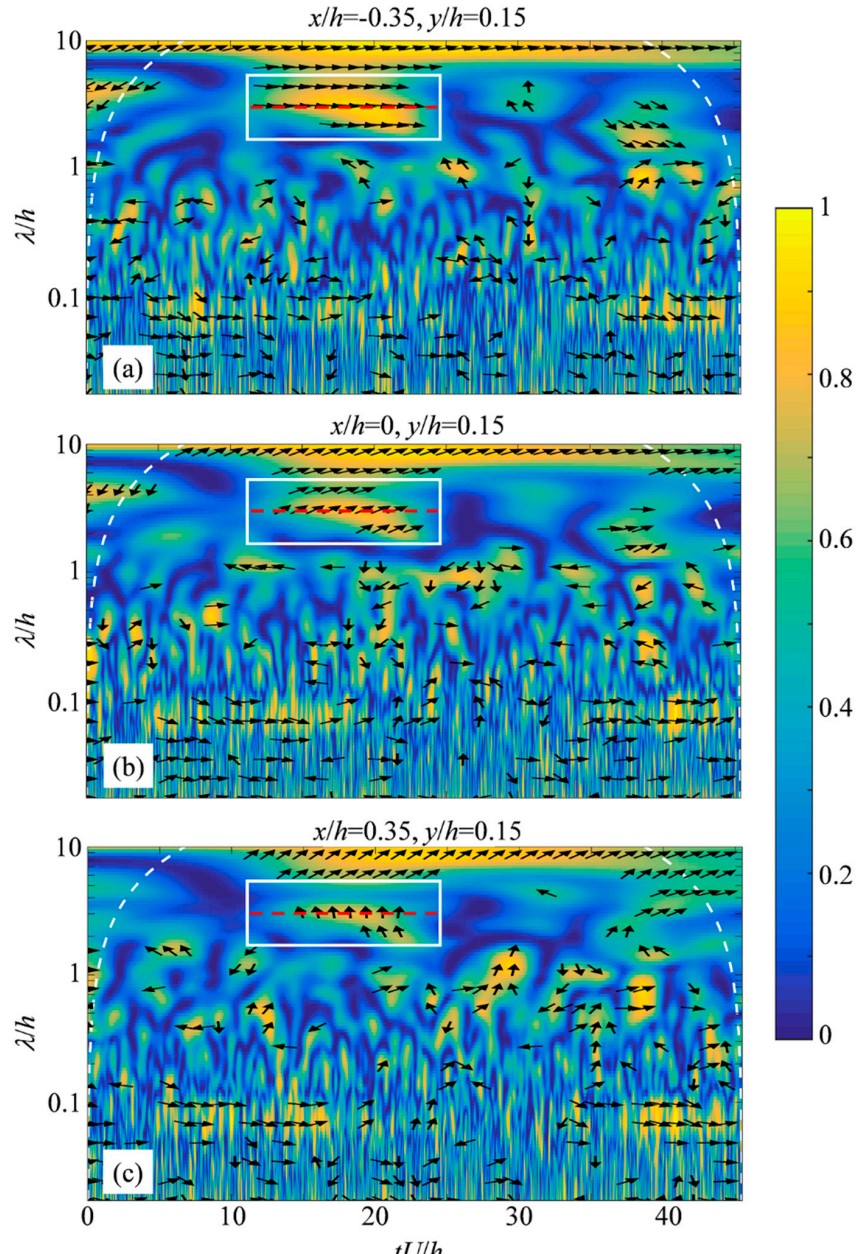

**Figure 5.** Wavelet coherent coefficients between the streamwise velocity series at different measurement points. The fixed point is ($x/h = 0$, $y/h = 0.5$), the moving points are ($x/h = -0.35$, $y/h = 0.15$), ($x/h = 0$, $y/h = 0.15$), and ($x/h = 0.35$, $y/h = 0.15$). The highly coherent area reduces and the phase angle increases when the lower point moving from the upstream to downstream of the fixed upper point.

The comparison of the coherent area during $11 < tU/h < 21$ between three points in Figure 5a–c shows the coherent area reduces when the measurement point moves from the upstream to downstream. It means the overlap time duration of the $3h$ scale motions between fixed and moving points reduces. This is caused by the incline feature of the $3h$ scale motions. Figure 6 sketches cartoons to show the explanation. Two solid red dots represent two measurement points. Blue and yellow shapes stand for the inclined $3h$ scale structures. The blue and yellow shape mark the starting and end time instant, respectively, that both measurement points are inside the structure. $D$ is the distance between the centers of the blue and yellow shapes. Thus $D/U$ is the time interval of both measurement points located in the structure, which is the highly coherent area in Figure 5. It can be seen from Figure 6 that $D$ is the greatest when the lower point is located at the upstream of the upper point (corresponding to

Figure 5a). *D* decreases when the lower point moves from the upstream to downstream, as shown in Figure 5b,c, because of the incline feature of the hairpin packets. In addition, the phase angle increases when the lower point moves from the upstream to downstream. This is also cased by the incline feature of the hairpin packets. When the lower point is located at the proper location of the upstream of the upper point, the two points almost enter the structure at the same time, as shown in Figure 6a. Thus, the typical signal of the hairpin packet presents at the same time in the velocity series in the two points and the phase difference is small. When the two points are on the vertical line, as shown in Figure 6b, the lower point enters the structure after the upper one enters the structure for a while. Therefore, there is hysteresis between the phases of the signal of the corresponding scale. As the lower point keeps on moving downstream, the phase difference will increase.

From the above discussion, the wavelet coherency analysis between the streamwise and wall-normal velocity series at the same point and the streamwise velocity series from different points not only detects the occurrence and scales of hairpin packets in open channel flows, but also reveal the internal velocity organizations and the tilt geometry of hairpin packets. The wavelet coherent coefficient and the phase angles present a powerful tool for the detection of the energetic coherent structures and the analysis of their internal organizations.

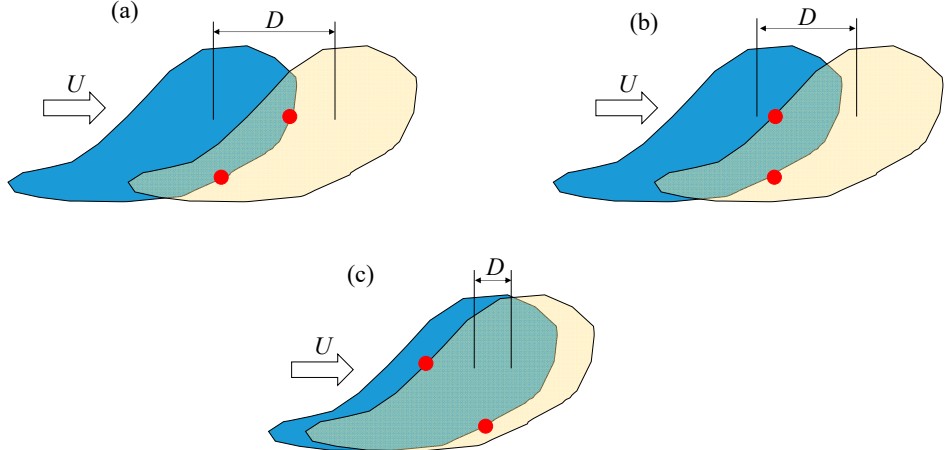

**Figure 6.** The explanation of the reducing wavelet coherent area and the increasing phase angle when the lower point moving from the upstream to downstream of the fixed upper point. When the lower point is located at the proper location of the upstream of the upper point, the two points almost enter the structure at the same time, as shown in (**a**). Thus, the typical signal of the hairpin packet presents at the same time in the velocity series in the two points and the phase difference is small, as shown in Figure 5a. When the two points are on the vertical line as shown in (**b**), the lower point enters the structure after the upper one enters the structure for a while. There is hysteresis between the phases of the signal of the corresponding scale (see Figure 5b). As the lower point keeps on moving downstream, the phase difference will increase, as in (**c**).

## 5. Concluding Remarks

Many studies based on single-point measurement data have demonstrated the impressive ability of the wavelet coherency analysis to catch the coherent structures in the wall-bounded flows, however, the question that how the events found by the wavelet coherency analysis based on single point measurement data relate to the features of the three-dimensional coherent structures remains open. Present study employed the TR-PIV system to produce a time series of a two dimensional flow field in the streamwise-wall-normal plane of steady open channel flows, which makes it possible to find out entire flow structures from the 2D flow field of the corresponding wavelet coherent events. Wavelet coherency analysis was applied on the velocity series from a single or multi-point to find the highly energetic and coherent events. The fluctuating velocity fields during the time period with high energy and high wavelet coherence were pieced together based on Taylor's frozen turbulence hypothesis to

explore the corresponding entire coherent structures of the wavelet events. The major finding of this study are as follows:

(1) The high value peaks in the pre-multiplied wavelet power spectrum curves stand for the energetic scales in the signal, and the high value areas in the local wavelet spectrum gives both the scales and the time instants of energetic motions. The $3h$ motion is the main energetic scale in open channel flows based on the wavelet spectral analysis on the TR-PIV data.

(2) The high wavelet coherent coefficients area from the wavelet coherency analysis can also detect the scale and the occurrence time instants of energetic motions, and the phase angles reveal the inner structure of the energetic motions.

(3) The wavelet coherency phase angle of $3h$ motions between the streamwise and vertical velocity of the same measurement point is $\pi$, which reveals that the velocity structure of these $3h$ motions is organized by the Q2 and Q4 events. By the wavelet coherency analysis between the streamwise velocities from different measurements points, the highly coherent area reduces and the phase angle increases when the lower point moving from the upstream to downstream of the fixed upper point, which reflects the $3h$ scale motions, is an incline structure.

(4) All the features of this $3h$ motions revealed by the wavelet coherency analysis support the hairpin packets model in open channel flows.

**Author Contributions:** Conceptualization, Q.Z.; methodology, K.C. and Y.Z; data curation, Y.Z.; writing—original draft preparation, K.C.; Y.Z., writing—review and editing, Q.Z.; supervision, Q.Z.; funding acquisition, Q.Z.

**Funding:** The study was supported by the National Natural Science Foundation of China (Grant No.51809268) and the Fundamental Research Funds for the Central Universities (China Agricultural University, Project No. 2019TC043).

**Conflicts of Interest:** The authors declare no conflict of interest.

## Symbol Table

| Symbol | Meaning | Symbol | Meaning |
|---|---|---|---|
| $u$ | streamwise fluctuation | $v$ | vertical fluctuation |
| $U$ | mean velocity | $\psi$ | wavelet function |
| $W_u$ | wavelet coefficient | $W_{uv}$ | cross wavelet coefficient |
| $E_{uu}$ | spectrum | $R_{uv}$ | correlation coefficient |
| $f$ | Fourier frequency | $k$ | wave number |
| $\lambda$ | wavelength | $\omega_0$ | wavelet center frequency |
| $\Re[\cdot]$ | real part of complex variables | $\Im[\cdot]$ | imaginary part of the complex variables |
| $J$ | bed slope | $h$ | water depth |
| $u_*$ | friction velocity | $Fr$ | Froude number |
| $Re$ | Reynolds number | $Re_\tau$ | friction Reynolds number |

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
