# Peer review of "Wavelet Coherency Structure in Open Channel Flow"

_water, doi:10.3390/w11081664_

Round 1

Reviewer 1 Report

This paper presents an application of wavelet coherency analysis to open channel flow. The authors compute the wavelet energy spectrum for a set of velocity measurements in shallow channel flow, and argue that an analysis based on wavelet coherency between the time series of velocity fluctuations in stream-wise and wall-normal directions yields further insight into the flow coherent structures observed.

I think the paper is interesting, but I found the analysis to be quite arcane. Maybe an expert in wavelet analysis can relate patterns in wavelength coherent coefficients to flow patterns, but I don’t think I would be able to do it based on this paper. For example, the authors relate the appearance of Figure 3 to 3h-scale structures continuously passing through the measurement point. Following the same argument, there would be a significant number of coherent with other lengths and at different times (i.e. the large yellow region near the upper corner of the plot). Why would one choose one family of structures over the others? And, perhaps more importantly, why would this be important at all? In the end, we need the actual flow measurements to plot the wavelet coherent coefficients, so it is not obvious that Figures 3 and 5 have information that is not already in Figure 2.

My suggestion is that the authors rewrite the Results section in a way that is: 1) More accessible to non-specialists, and 2) more quantitative. As written, it seems that only experienced researchers would be able to extract any information from these wavelength coherency plots. If one were to identify these patterns automatically using some pattern detection algorithm, what would the algorithm be looking for?

Minor comments:

- Math symbols missing in the main text: for example lines 81,83,87,92,95,225,228,234

- Line 189: “filed”

- Define what events Q2 and Q4 are.

- Use Figure instead of figure.

- Lines 225 and 226 need to be rewritten. Maybe “highly coherent,” “continuously passing…” The phrase “The most portion of the duration consist of the hairpin packets when checking from figure 2” needs to be rewritten.

Author Response

Point 1: This paper presents an application of wavelet coherency analysis to open channel flow. The authors compute the wavelet energy spectrum for a set of velocity measurements in shallow channel flow, and argue that an analysis based on wavelet coherency between the time series of velocity fluctuations in stream-wise and wall-normal directions yields further insight into the flow coherent structures observed.

    I think the paper is interesting, but I found the analysis to be quite arcane. Maybe an expert in wavelet analysis can relate patterns in wavelength coherent coefficients to flow patterns, but I don’t think I would be able to do it based on this paper.

Response 1: Thank you very much for your valuable comments, which help us further improve this paper. Wavelet coherency is a relatively new approach applied in flow analysis. The purpose of this paper is to reveal two dimensional structures of the high wavelet coherence events in the single-point data analysis in the turbulence. So we focus on coherence analysis rather than on the physical meaning of wavelet coherence itself. We used the MATLAB function “wcoherence” to compute the wavelet coherency between two signals in this study. One can find the wavelet coherency of analytical signals in the documentation of the MATLAB function.

Obtain the wavelet coherence data for two signals, specifying a sampling interval of 0.001. Both X and Y signals (see Figure 1) consist of two sine waves (10 Hz and 75 Hz) in white noise. The overlapping section for 10Hz is 1.2<t<3, and for 75Hz is 0.4<t<4.4, and there is a quarter phase delay between X and Y for both frequency. From there wavelet coherency we can see that two high wavelet coherence areas correspond strictly to the overlapping sections, and the wavelet coherency analysis reveals the phase delay accurately (horizontal and right arrow indicates zeros phase difference, and vertical up ones indicates p/2 phase difference). Then the phase difference was adjusted between X and Y. The phase delay for 10Hz is set to 3p/4, and for 75Hz is p/4. It can be seen clearly in Figure 2 that the new phase delay is revealed accurately..

Based on the reviewer's suggestion, we added a paragraph in the end of section 2. This paragraph describes the MATLAB function we used and briefly introduces the examples for the readers who want to visualize the physical meaning of the wavelet coherency.

Figure 1. X and Y signals and the wavelet coherency

Figure 2. Wavelet coherency for the adjusted signals

Point 2: For example, the authors relate the appearance of Figure 3 to 3h-scale structures continuously passing through the measurement point. Following the same argument, there would be a significant number of coherent with other lengths and at different times (i.e. the large yellow region near the upper corner of the plot). Why would one choose one family of structures over the others? And, perhaps more importantly, why would this be important at all?

Response 2: We agree with the reviewer’s viewpoint. There are different scales of coherent structures in open channel turbulence. From our paper, we can see that the scale of coherent structure can be from 0.01h to tens times of water depth. From Figure 3 in paper, the most remarkably coherence appears in 13 < tU/h <32 (marked by the white rectangle) near the scale 3h (marked by the red dash line) and in tU/h >35 at the scale 3h to 10h (the upper right corner). Most of the coherence area of tU/h >35 is beyond the white dash line, which is the cone of influence for the wavelet. The wavelet coherent value is untrusted because they are too close to the right boundary of data. Thus, we only focus on the structure in the white box. We added a explanation in section 4.2.1 in the revised version.

Point 3: In the end, we need the actual flow measurements to plot the wavelet coherent coefficients, so it is not obvious that Figures 3 and 5 have information that is not already in Figure 2.

Response 3: The wavelet coherency is mainly a method for single or multiple point measurement data analysis, and all results shown in figure 1, 3 and 5 are based on the single or two point velocity time series extract from the 2D PIV velocity time series. Although we can easily get two- or even three-dimensional flow fields in the laboratory today, a large number of tasks in the actual application scenario are still completed by single-point measurement. Therefore, it is instructive for the data analysis to explore the three-dimensional coherent structure corresponding to events in single-point measurement data. Present study employs the time-resolved particle imaging velocimetry system (TR-PIV) to get the two dimensional flow field time series, which makes it possible to find out the flow structures from the 2D flow fields at the corresponding wavelet coherent events from the single point time series extract from PIV data. In addition, the original flow field does contain all the information about the flow, but we need to use effective statistical methods to extract the important events usually happening in the flow. This study shows that the wavelet coherency is a good analysis method. We added some explanations in section 1 in the revised version.

Point 4: My suggestion is that the authors rewrite the Results section in a way that is: 1) More accessible to non-specialists, and 2) more quantitative. As written, it seems that only experienced researchers would be able to extract any information from these wavelength coherency plots. If one were to identify these patterns automatically using some pattern detection algorithm, what would the algorithm be looking for?

Response 4: Based on the reviewer's suggestion, we rewrite the concluding remarks. The significance of various output parameters of wavelet coherency analysis are clearly stated for the non-specialists. One can design the pattern detection algorithm easily based on these understanding of wavelet coherency parameters if he want to identify energetic flow patterns automatically.

Point 5: Minor comments:

- Math symbols missing in the main text: for example lines 81,83,87,92,95,225,228,234

- Line 189: “filed”

- Define what events Q2 and Q4 are.

- Use Figure instead of figure.

- Lines 225 and 226 need to be rewritten. Maybe “highly coherent,” “continuously passing…” The phrase “The most portion of the duration consist of the hairpin packets when checking from figure 2” needs to be rewritten.

Response 5: We have corrected these sentences in the revised version based on the reviewer's comments.

Reviewer 2 Report

The reviewer wants to thank the authors for their paper about the wavelet analysis of turbulence in an open channel flow. He/she has some (small) points and questions to ask:

1)   Line (L) 4 and 7 missing email 

2)   Abstract and L71 TR-PIV is used twice before it is fully explained in L137. Please don’t use acronyms without full description in the abstract and introduce each one first. 

3)   There are some formatting issues with special characters in L83, 87, 92, Table 1 including caption, 225, 228 (two), 234 

4)   Section 3 starting L122: It is not fully clear, in which direction the laser cuts through the water and where the camera is located. Please just add here some very basic information, which allows to fully understand the later following results. 

5)   General: Please expand the captions to the figures hence it is possible to fully understand it without reading the text. Also add the flow direction, where is is sensible.

6)   Figure 4 left the second axis is not named. 

7)   An additional notation/nomenclature section would be very useful. 

The reviewer is looking forward to read the paper again in detail. 

Author Response

Point 1: The reviewer wants to thank the authors for their paper about the wavelet analysis of turbulence in an open channel flow. He/she has some (small) points and questions to ask:

1)   Line (L) 4 and 7 missing email

2)   Abstract and L71 TR-PIV is used twice before it is fully explained in L137. Please don’t use acronyms without full description in the abstract and introduce each one first.

3)   There are some formatting issues with special characters in L83, 87, 92, Table 1 including caption, 225, 228 (two), 234. 

5)   General: Please expand the captions to the figures hence it is possible to fully understand it without reading the text. Also add the flow direction, where is is sensible.

6)   Figure 4 left the second axis is not named.

Response 1: We have corrected these sentences in the revised version based on the reviewer's comments.

Point 2: 4)   Section 3 starting L122: It is not fully clear, in which direction the laser cuts through the water and where the camera is located. Please just add here some very basic information, which allows to fully understand the later following results.

Response 2: The laser sheet was projected from the channel bed and it was located at the central line of the channel. The camera was set at the side of the channel and the laser sheet plane and the CMOS plane of the camera kept parallel to the mid-vertical plane of the channel. We added some explanations in the revised version.

Point 3: 7)   An additional notation/nomenclature section would be very useful.

Response 3: A symbol table is added as Appendix A in the revised version.

Round 2

Reviewer 1 Report

The authors have addressed my previous comments. I recommend publication.

Reviewer 2 Report

Looking forward to the publication. Thank you.